# A-NICE-MC: Adversarial Training for MCMC

**Jiaming Song**
Stanford University
tsong@cs.stanford.edu

**Shengjia Zhao**
Stanford University
zhaosj12@cs.stanford.edu

**Stefano Ermon**
Stanford University
ermon@cs.stanford.edu

## Abstract

Existing Markov Chain Monte Carlo (MCMC) methods are either based on general-purpose and domain-agnostic schemes, which can lead to slow convergence, or problem-specific proposals hand-crafted by an expert. In this paper, we propose A-NICE-MC, a novel method to automatically design efficient Markov chain kernels tailored for a specific domain. First, we propose an efficient likelihood-free adversarial training method to train a Markov chain and mimic a given data distribution. Then, we leverage flexible volume preserving flows to obtain parametric kernels for MCMC. Using a bootstrap approach, we show how to train efficient Markov chains to sample from a prescribed posterior distribution by iteratively improving the quality of both the model and the samples. Empirical results demonstrate that A-NICE-MC combines the strong guarantees of MCMC with the expressiveness of deep neural networks, and is able to significantly outperform competing methods such as Hamiltonian Monte Carlo.

## 1 Introduction

Variational inference (VI) and Monte Carlo (MC) methods are two key approaches to deal with complex probability distributions in machine learning. The former approximates an intractable distribution by solving a variational optimization problem to minimize a divergence measure with respect to some tractable family. The latter approximates a complex distribution using a small number of typical states, obtained by sampling ancestrally from a proposal distribution or iteratively using a suitable Markov chain (Markov Chain Monte Carlo, or MCMC).

Recent progress in deep learning has vastly advanced the field of variational inference. Notable examples include black-box variational inference and variational autoencoders [1–3], which enabled variational methods to benefit from the expressive power of deep neural networks, and adversarial training [4, 5], which allowed the training of new families of implicit generative models with efficient ancestral sampling. MCMC methods, on the other hand, have not benefited as much from these recent advancements. Unlike variational approaches, MCMC methods are iterative in nature and do not naturally lend themselves to the use of expressive function approximators [6, 7]. Even evaluating an existing MCMC technique is often challenging, and natural performance metrics are intractable to compute [8–11]. Defining an objective to improve the performance of MCMC that can be easily optimized in practice over a large parameter space is itself a difficult problem [12, 13].

To address these limitations, we introduce A-NICE-MC, a new method for training flexible MCMC kernels, e.g., parameterized using (deep) neural networks. Given a kernel, we view the resulting Markov Chain as an implicit generative model, i.e., one where sampling is efficient but evaluating the (marginal) likelihood is intractable. We then propose adversarial training as an effective, likelihood-free method for training a Markov chain to match a target distribution. First, we show it can be used in a learning setting to directly approximate an (empirical) data distribution. We then use the approach to train a Markov Chain to sample efficiently from a model prescribed by an analytic expression (e.g., a Bayesian posterior distribution), the classic use case for MCMC techniques. We leverage flexible volume preserving flow models [14] and a "bootstrap" technique to automatically design powerful

domain-specific proposals that combine the guarantees of MCMC and the expressiveness of neural networks. Finally, we propose a method that decreases autocorrelation and increases the effective sample size of the chain as training proceeds. We demonstrate that these trained operators are able to significantly outperform traditional ones, such as Hamiltonian Monte Carlo, in various domains.

## 2  Notations and Problem Setup

A sequence of continuous random variables $\{x_t\}_{t=0}^{\infty}$, $x_t \in \mathbb{R}^n$, is drawn through the following Markov chain:

$$x_0 \sim \pi^0 \qquad x_{t+1} \sim T_\theta(x_{t+1}|x_t)$$

where $T_\theta(\cdot|x)$ is a time-homogeneous stochastic transition kernel parametrized by $\theta \in \Theta$ and $\pi^0$ is some initial distribution for $x_0$. In particular, we assume that $T_\theta$ is defined through an implicit generative model $f_\theta(\cdot|x, v)$, where $v \sim p(v)$ is an auxiliary random variable, and $f_\theta$ is a deterministic transformation (e.g., a neural network). Let $\pi_\theta^t$ denote the distribution for $x_t$. If the Markov chain is both irreducible and positive recurrent, then it has an unique stationary distribution $\pi_\theta = \lim_{t \to \infty} \pi_\theta^t$. We assume that this is the case for all the parameters $\theta \in \Theta$.

Let $p_d(x)$ be a target distribution over $x \in \mathbb{R}^n$, e.g, a data distribution or an (intractable) posterior distribution in a Bayesian inference setting. Our objective is to find a $T_\theta$ such that:

1. **Low bias:** The stationary distribution is close to the target distribution (minimize $|\pi_\theta - p_d|$).
2. **Efficiency:** $\{\pi_\theta^t\}_{t=0}^{\infty}$ converges quickly (minimize $t$ such that $|\pi_\theta^t - p_d| < \delta$).
3. **Low variance:** Samples from one chain $\{x_t\}_{t=0}^{\infty}$ should be as uncorrelated as possible (minimize autocorrelation of $\{x_t\}_{t=0}^{\infty}$).

We think of $\pi_\theta$ as a stochastic generative model, which can be used to efficiently produce samples with certain characteristics (specified by $p_d$), allowing for efficient Monte Carlo estimates. We consider two settings for specifying the target distribution. The first is a *learning* setting where we do not have an analytic expression for $p_d(x)$ but we have access to typical samples $\{s_i\}_{i=1}^{m} \sim p_d$; in the second case we have an analytic expression for $p_d(x)$, possibly up to a normalization constant, but no access to samples. The two cases are discussed in Sections 3 and 4 respectively.

## 3  Adversarial Training for Markov Chains

Consider the setting where we have direct access to samples from $p_d(x)$. Assume that the transition kernel $T_\theta(x_{t+1}|x_t)$ is the following implicit generative model:

$$v \sim p(v) \quad x_{t+1} = f_\theta(x_t, v) \tag{1}$$

Assuming a stationary distribution $\pi_\theta(x)$ exists, the value of $\pi_\theta(x)$ is typically intractable to compute. The marginal distribution $\pi_\theta^t(x)$ at time $t$ is also intractable, since it involves integration over all the possible paths (of length $t$) to $x$. However, we can directly obtain samples from $\pi_\theta^t$, which will be close to $\pi_\theta$ if $t$ is large enough (assuming ergodicity). This aligns well with the idea of generative adversarial networks (GANs), a likelihood free method which only requires samples from the model.

Generative Adversarial Network (GAN) [4] is a framework for training deep generative models using a two player minimax game. A generator network $G$ generates samples by transforming a noise variable $z \sim p(z)$ into $G(z)$. A discriminator network $D(\mathbf{x})$ is trained to distinguish between "fake" samples from the generator and "real" samples from a given data distribution $p_d$. Formally, this defines the following objective (Wasserstein GAN, from [15])

$$\min_G \max_D V(D, G) = \min_G \max_D \mathbb{E}_{x \sim p_d}[D(x)] - \mathbb{E}_{z \sim p(z)}[D(G(z))] \tag{2}$$

In our setting, we could assume $p_d(x)$ is the empirical distribution from the samples, and choose $z \sim \pi^0$ and let $G_\theta(z)$ be the state of the Markov Chain after $t$ steps, which is a good approximation of $\pi_\theta$ if $t$ is large enough. However, optimization is difficult because we do not know a reasonable $t$ in advance, and the gradient updates are expensive due to backpropagation through the entire chain.

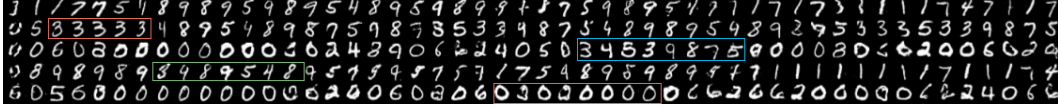

Figure 1: Visualizing samples of $\pi_1$ to $\pi_{50}$ (each row) from a model trained on the MNIST dataset. Consecutive samples can be related in label (red box), inclination (green box) or width (blue box).

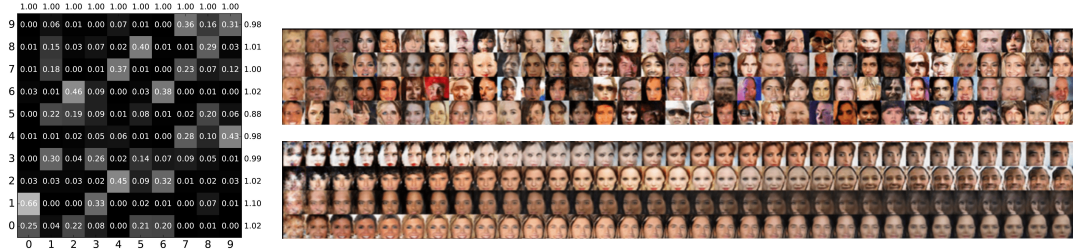

Figure 2: $T_\theta(y_{t+1}|y_t)$.    Figure 3: Samples of $\pi_1$ to $\pi_{30}$ from models (top: without shortcut connections; bottom: with shortcut connections) trained on the CelebA dataset.

Therefore, we propose a more efficient approximation, called *Markov GAN* (MGAN):

$$\min_\theta \max_D \mathbb{E}_{x \sim p_d}[D(x)] - \lambda \mathbb{E}_{\bar{x} \sim \pi_\theta^b}[D(\bar{x})] - (1-\lambda)\mathbb{E}_{x_d \sim p_d, \bar{x} \sim T_\theta^m(\bar{x}|x_d)}[D(\bar{x})] \quad (3)$$

where $\lambda \in (0,1), b \in \mathbb{N}^+, m \in \mathbb{N}^+$ are hyperparameters, $\bar{x}$ denotes "fake" samples from the generator and $T_\theta^m(x|x_d)$ denotes the distribution of $x$ when the transition kernel is applied $m$ times, starting from some "real" sample $x_d$.

We use two types of samples from the generator for training, optimizing $\theta$ such that the samples will fool the discriminator:

1. Samples obtained after $b$ transitions $\bar{x} \sim \pi_\theta^b$, starting from $x_0 \sim \pi^0$.
2. Samples obtained after $m$ transitions, starting from a data sample $x_d \sim p_d$.

Intuitively, the first condition encourages the Markov Chain to converge towards $p_d$ over relatively short runs (of length $b$). The second condition enforces that $p_d$ is a fixed point for the transition operator. [1] Instead of simulating the chain until convergence, which will be especially time-consuming if the initial Markov chain takes many steps to mix, the generator would run only $(b+m)/2$ steps on average. Empirically, we observe better training times by uniformly sampling $b$ from $[1, B]$ and $m$ from $[1, M]$ respectively in each iteration, so we use $B$ and $M$ as the hyperparameters for our experiments.

### 3.1  Example: Generative Model for Images

We experiment with a distribution $p_d$ over images, such as digits (MNIST) and faces (CelebA). In the experiments, we parametrize $f_\theta$ to have an autoencoding structure, where the auxiliary variable $v \sim \mathcal{N}(0, I)$ is directly added to the latent code of the network serving as a source of randomness:

$$z = \text{encoder}_\theta(x_t) \quad z' = \text{ReLU}(z + \beta v) \quad x_{t+1} = \text{decoder}_\theta(z') \quad (4)$$

where $\beta$ is a hyperparameter we set to $0.1$. While sampling is inexpensive, evaluating probabilities according to $T_\theta(\cdot|x_t)$ is generally intractable as it would require integration over $v$. The starting distribution $\pi_0$ is a factored Gaussian distribution with mean and standard deviation being the mean and standard deviation of the training set. We include all the details, which ares based on the DCGAN [16] architecture, in Appendix E.1. All the models are trained with the gradient penalty objective for Wasserstein GANs [17, 15], where $\lambda = 1/3$, $B = 4$ and $M = 3$.

We visualize the samples generated from our trained Markov chain in Figures 1 and 3, where each row shows consecutive samples from the same chain (we include more images in Appendix F) From

Figure 1 it is clear that $x_{t+1}$ is related to $x_t$ in terms of high-level properties such as digit identity (label). Our model learns to find and "move between the modes" of the dataset, instead of generating a single sample ancestrally. This is drastically different from other iterative generative models trained with maximum likelihood, such as Generative Stochastic Networks (GSN, [18]) and Infusion Training (IF, [19]), because when we train $T_\theta(x_{t+1}|x_t)$ we are not specifying a particular target for $x_{t+1}$. In fact, to maximize the discriminator score the model (generator) may choose to generate some $x_{t+1}$ near a different mode.

To further investigate the frequency of various modes in the stationary distribution, we consider the class-to-class transition probabilities for MNIST. We run one step of the transition operator starting from real samples where we have class labels $y \in \{0, \ldots, 9\}$, and classify the generated samples with a CNN. We are thus able to quantify the transition matrix for labels in Figure 2. Results show that class probabilities are fairly uniform and range between $0.09$ and $0.11$.

Although it seems that the MGAN objective encourages rapid transitions between different modes, it is not always the case. In particular, as shown in Figure 3, adding residual connections [20] and highway connections [21] to an existing model can significantly increase the time needed to transition between modes. This suggests that the time needed to transition between modes can be affected by the architecture we choose for $f_\theta(x_t, v)$. If the architecture introduces an information bottleneck which forces the model to "forget" $x_t$, then $x_{t+1}$ will have higher chance to occur in another mode; on the other hand, if the model has shortcut connections, it tends to generate $x_{t+1}$ that are close to $x_t$. The increase in autocorrelation will hinder performance if samples are used for Monte Carlo estimates.

## 4 Adversarial Training for Markov Chain Monte Carlo

We now consider the setting where the target distribution $p_d$ is specified by an analytic expression:

$$p_d(x) \propto \exp(-U(x)) \tag{5}$$

where $U(x)$ is a known "energy function" and the normalization constant in Equation (5) might be intractable to compute. This form is very common in Bayesian statistics [22], computational physics [23] and graphics [24]. Compared to the setting in Section 3, there are two additional challenges:

1. We want to train a Markov chain such that the stationary distribution $\pi_\theta$ is *exactly* $p_d$;
2. We do not have direct access to samples from $p_d$ during training.

### 4.1 Exact Sampling Through MCMC

We use ideas from the Markov Chain Monte Carlo (MCMC) literature to satisfy the first condition and guarantee that $\{\pi_\theta^t\}_{t=0}^\infty$ will asymptotically converge to $p_d$. Specifically, we require the transition operator $T_\theta(\cdot|x)$ to satisfy the *detailed balance* condition:

$$p_d(x)T_\theta(x'|x) = p_d(x')T_\theta(x|x') \tag{6}$$

for all $x$ and $x'$. This condition can be satisfied using Metropolis-Hastings (MH), where a sample $x'$ is first obtained from a *proposal distribution* $g_\theta(x'|x)$ and accepted with the following probability:

$$A_\theta(x'|x) = \min\left(1, \frac{p_d(x')}{p_d(x)}\frac{g_\theta(x|x')}{g_\theta(x'|x)}\right) = \min\left(1, \exp(U(x) - U(x'))\frac{g_\theta(x|x')}{g_\theta(x'|x)}\right) \tag{7}$$

Therefore, the resulting MH transition kernel can be expressed as $T_\theta(x'|x) = g_\theta(x'|x)A_\theta(x'|x)$ (if $x \neq x'$), and it can be shown that $p_d$ is stationary for $T_\theta(\cdot|x)$ [25].

The idea is then to optimize for a good proposal $g_\theta(x'|x)$. We can set $g_\theta$ directly as in Equation (1) (if $f_\theta$ takes a form where the probability $g_\theta$ can be computed efficiently), and attempt to optimize the MGAN objective in Eq. (3) (assuming we have access to samples from $p_d$, a challenge we will address later). Unfortunately, Eq. (7) is not differentiable - the setting is similar to policy gradient optimization in reinforcement learning. In principle, score function gradient estimators (such as REINFORCE [26]) could be used in this case; in our experiments, however, this approach leads to extremely low acceptance rates. This is because during initialization, the ratio $g_\theta(x|x')/g_\theta(x'|x)$ can be extremely low, which leads to low acceptance rates and trajectories that are not informative for training. While it might be possible to optimize directly using more sophisticated techniques from the RL literature, we introduce an alternative approach based on volume preserving dynamics.

## 4.2 Hamiltonian Monte Carlo and Volume Preserving Flow

To gain some intuition to our method, we introduce Hamiltonian Monte Carlo (HMC) and volume preserving flow models [27]. HMC is a widely applicable MCMC method that introduces an auxiliary "velocity" variable $v$ to $g_\theta(x'|x)$. The proposal first draws $v$ from $p(v)$ (typically a factored Gaussian distribution) and then obtains $(x', v')$ by simulating the dynamics (and inverting $v$ at the end of the simulation) corresponding to the Hamiltonian

$$H(x, v) = v^\top v/2 + U(x) \tag{8}$$

where $x$ and $v$ are iteratively updated using the *leapfrog integrator* (see [27]). The transition from $(x, v)$ to $(x', v')$ is deterministic, invertible and volume preserving, which means that

$$g_\theta(x', v'|x, v) = g_\theta(x, v|x', v') \tag{9}$$

MH acceptance (7) is computed using the distribution $p(x, v) = p_d(x)p(v)$, where the acceptance probability is $p(x', v')/p(x, v)$ since $g_\theta(x', v'|x, v)/g_\theta(x, v|x', v') = 1$. We can safely discard $v'$ after the transition since $x$ and $v$ are independent.

Let us return to the case where the proposal is parametrized by a neural network; if we could satisfy Equation 9 then we could significantly improve the acceptance rate compared to the "REINFORCE" setting. Fortunately, we can design such an proposal by using a volume preserving flow model [14].

A flow model [14, 28–30] defines a generative model for $x \in \mathbb{R}^n$ through a bijection $f : h \to x$, where $h \in \mathbb{R}^n$ have the same number of dimensions as $x$ with a fixed prior $p_H(h)$ (typically a factored Gaussian). In this form, $p_X(x)$ is tractable because

$$p_X(x) = p_H(f^{-1}(x)) \left| \det \frac{\partial f^{-1}(x)}{\partial x} \right|^{-1} \tag{10}$$

and can be optimized by maximum likelihood.

In the case of a *volume preserving flow model* $f$, the determinant of the Jacobian $\frac{\partial f(h)}{\partial h}$ is one. Such models can be constructed using *additive coupling layers*, which first partition the input into two parts, $y$ and $z$, and then define a mapping from $(y, z)$ to $(y', z')$ as:

$$y' = y \qquad z' = z + m(y) \tag{11}$$

where $m(\cdot)$ can be a complex function. By stacking multiple coupling layers the model becomes highly expressive. Moreover, once we have the forward transformation $f$, the backward transformation $f^{-1}$ can be easily derived. This family of models are called *Non-linear Independent Components Estimation* (NICE)[14].

## 4.3 A NICE Proposal

HMC has two crucial components. One is the introduction of the auxiliary variable $v$, which prevents random walk behavior; the other is the symmetric proposal in Equation (9), which allows the MH step to only consider $p(x, v)$. In particular, if we simulate the Hamiltonian dynamics (the deterministic part of the proposal) twice starting from any $(x, v)$ (without MH or resampling $v$), we will always return to $(x, v)$.

Auxiliary variables can be easily integrated into neural network proposals. However, it is hard to obtain symmetric behavior. If our proposal is deterministic, then $f_\theta(f_\theta(x, v)) = (x, v)$ should hold for all $(x, v)$, a condition which is difficult to achieve [2]. Therefore, we introduce a proposal which satisfies Equation (9) for any $\theta$, while preventing random walk in practice by resampling $v$ after every MH step.

Our proposal considers a NICE model $f_\theta(x, v)$ with its inverse $f_\theta^{-1}$, where $v \sim p(v)$ is the auxiliary variable. We draw a sample $x'$ from the proposal $g_\theta(x', v'|x, v)$ using the following procedure:

1. Randomly sample $v \sim p(v)$ and $u \sim \text{Uniform}[0, 1]$;
2. If $u > 0.5$, then $(x', v') = f_\theta(x, v)$;

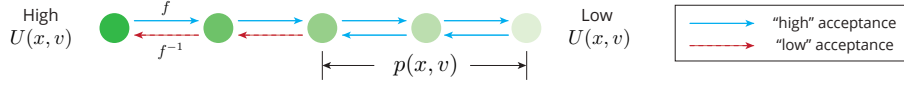

Figure 4: Sampling process of A-NICE-MC. Each step, the proposal executes $f_\theta$ or $f_\theta^{-1}$. Outside the high probability regions $f_\theta$ will guide $x$ towards $p_d(x)$, while MH will tend to reject $f_\theta^{-1}$. Inside high probability regions both operations will have a reasonable probability of being accepted.

3. If $u \le 0.5$, then $(x', v') = f_\theta^{-1}(x, v)$.

We call this proposal a *NICE proposal* and introduce the following theorem.

**Theorem 1.** *For any $(x, v)$ and $(x', v')$ in their domain, a NICE proposal $g_\theta$ satisfies*

$$g_\theta(x', v'|x, v) = g_\theta(x, v|x', v')$$

*Proof.* In Appendix C. □

### 4.4 Training A NICE Proposal

Given any NICE proposal with $f_\theta$, the MH acceptance step guarantees that $p_d$ is a stationary distribution, yet the ratio $p(x', v')/p(x, v)$ can still lead to low acceptance rates unless $\theta$ is carefully chosen. Intuitively, we would like to train our proposal $g_\theta$ to produce samples that are likely under $p(x, v)$.

Although the proposal itself is non-differentiable w.r.t. $x$ and $v$, we do not require score function gradient estimators to train it. In fact, if $f_\theta$ is a bijection between samples in high probability regions, then $f_\theta^{-1}$ is automatically also such a bijection. Therefore, we ignore $f_\theta^{-1}$ during training and only train $f_\theta(x, v)$ to reach the target distribution $p(x, v) = p_d(x)p(v)$. For $p_d(x)$, we use the MGAN objective in Equation (3); for $p(v)$, we minimize the distance between the distribution for the generated $v'$ (tractable through Equation (10)) and the prior distribution $p(v)$ (which is a factored Gaussian):

$$\min_\theta \max_D L(x; \theta, D) + \gamma L_d(p(v), p_\theta(v')) \tag{12}$$

where $L$ is the MGAN objective, $L_d$ is an objective that measures the divergence between two distributions and $\gamma$ is a parameter to balance between the two factors; in our experiments, we use KL divergence for $L_d$ and $\gamma = 1$ [3].

Our transition operator includes a trained NICE proposal followed by a Metropolis-Hastings step, and we call the resulting Markov chain *Adversarial NICE Monte Carlo* (A-NICE-MC). The sampling process is illustrated in Figure 4. Intuitively, if $(x, v)$ lies in a high probability region, then both $f_\theta$ and $f_\theta^{-1}$ should propose a state in another high probability region. If $(x, v)$ is in a low-probability probability region, then $f_\theta$ would move it closer to the target, while $f_\theta^{-1}$ does the opposite. However, the MH step will bias the process towards high probability regions, thereby suppressing the random-walk behavior.

### 4.5 Bootstrap

The main remaining challenge is that we do not have direct access to samples from $p_d$ in order to train $f_\theta$ according to the adversarial objective in Equation (12), whereas in the case of Section 3, we have a dataset to get samples from the data distribution.

In order to retrieve samples from $p_d$ and train our model, we use a bootstrap process [33] where the quality of samples used for adversarial training should increase over time. We obtain initial samples by running a (possibly) slow mixing operator $T_{\theta_0}$ with stationary distribution $p_d$ starting from an arbitrary initial distribution $\pi_0$. We use these samples to train our model $f_{\theta_i}$, and then use it to obtain new samples from our trained transition operator $T_{\theta_i}$; by repeating the process we can obtain samples of better quality which should in turn lead to a better model.

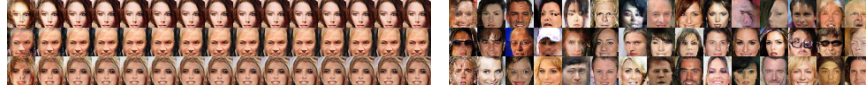

Figure 5: *Left*: Samples from a model with shortcut connections trained with ordinary discriminator. *Right*: Samples from the same model trained with a pairwise discriminator.



Figure 6: Densities of **ring**, **mog2**, **mog6** and **ring5** (from left to right).

### 4.6 Reducing Autocorrelation by Pairwise Discriminator

An important metric for evaluating MCMC algorithms is the effective sample size (ESS), which measures the number of "effective samples" we obtain from running the chain. As samples from MCMC methods are not i.i.d., to have higher ESS we would like the samples to be as independent as possible (low autocorrelation). In the case of training a NICE proposal, the objective in Equation (3) may lead to high autocorrelation even though the acceptance rate is reasonably high. This is because the coupling layer contains residual connections from the input to the output; as shown in Section 3.1, such models tend to learn an identity mapping and empirically they have high autocorrelation.

We propose to use a *pairwise discriminator* to reduce autocorrelation and improve ESS. Instead of scoring one sample at a time, the discriminator scores two samples $(x_1, x_2)$ at a time. For "real data" we draw two independent samples from our bootstrapped samples; for "fake data" we draw $x_2 \sim T_\theta^m(\cdot|x_1)$ such that $x_1$ is either drawn from the data distribution or from samples after running the chain for $b$ steps, and $x_2$ is the sample after running the chain for $m$ steps, which is similar to the samples drawn in the original MGAN objective.

The optimal solution would be match both distributions of $x_1$ and $x_2$ to the target distribution. Moreover, if $x_1$ and $x_2$ are correlated, then the discriminator should be able distinguish the "real" and "fake" pairs, so the model is forced to generate samples with little autocorrelation. More details are included in Appendix D. The pairwise discriminator is conceptually similar to the minibatch discrimination layer [34]; the difference is that we provide correlated samples as "fake" data, while [34] provides independent samples that might be similar.

To demonstrate the effectiveness of the pairwise discriminator, we show an example for the image domain in Figure 5, where the same model with shortcut connections is trained with and without pairwise discrimination (details in Appendix E.1); it is clear from the variety in the samples that the pairwise discriminator significantly reduces autocorrelation.

## 5 Experiments

Code for reproducing the experiments is available at https://github.com/ermongroup/a-nice-mc.

To demonstrate the effectiveness of A-NICE-MC, we first compare its performance with HMC on several synthetic 2D energy functions: **ring** (a ring-shaped density), **mog2** (a mixture of 2 Gaussians) **mog6** (a mixture of 6 Gaussians), **ring5** (a mixture of 5 distinct rings). The densities are illustrated in Figure 6 (Appendix E.2 has the analytic expressions). *ring* has a single connected component of high-probability regions and HMC performs well; *mog2*, *mog6* and *ring5* are selected to demonstrate cases where HMC fails to move across modes using gradient information. A-NICE-MC performs well in all the cases.

We use the same hyperparameters for all the experiments (see Appendix E.4 for details). In particular, we consider $f_\theta(x, v)$ with three coupling layers, which update $v$, $x$ and $v$ respectively. This is to ensure that both $x$ and $v$ could affect the updates to $x'$ and $v'$.

**How does A-NICE-MC perform?** We evaluate and compare ESS and ESS per second (ESS/s) for both methods in Table 1. For *ring*, *mog2*, *mog6*, we report the smallest ESS of all the dimensions

Table 1: Performance of MCMC samplers as measured by Effective Sample Size (ESS). Higher is better (1000 maximum). Averaged over 5 runs under different initializations. See Appendix A for the ESS formulation, and Appendix E.3 for how we benchmark the running time of both methods.

| ESS | A-NICE-MC | HMC | | ESS/s | A-NICE-MC | HMC |
|-----|-----------|-----|--|-------|-----------|-----|
| ring | **1000.00** | **1000.00** | | ring | **128205** | 121212 |
| mog2 | **355.39** | 1.00 | | mog2 | **50409** | 78 |
| mog6 | **320.03** | 1.00 | | mog6 | **40768** | 39 |
| ring5 | **155.57** | 0.43 | | ring5 | **19325** | 29 |

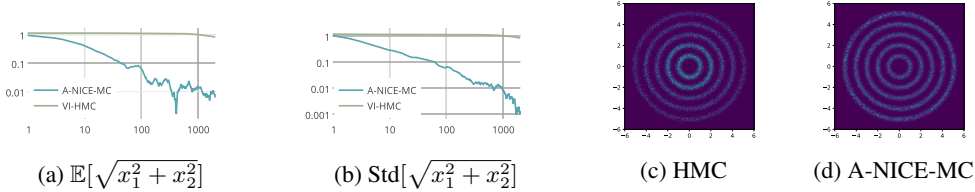

(a) $\mathbb{E}[\sqrt{x_1^2 + x_2^2}]$    (b) $\mathrm{Std}[\sqrt{x_1^2 + x_2^2}]$    (c) HMC    (d) A-NICE-MC

Figure 7: (a-b) Mean absolute error for estimating the statistics in *ring5* w.r.t. simulation length. Averaged over 100 chains. (c-d) Density plots for both methods. When the initial distribution is a Gaussian centered at the origin, HMC overestimates the densities of the rings towards the center.

(as in [35]); for *ring5*, we report the ESS of the distance between the sample and the origin, which indicates mixing across different rings. In the four scenarios, HMC performed well only in *ring*; in cases where modes are distant from each other, there is little gradient information for HMC to move between modes. On the other hand, A-NICE-MC is able to freely move between the modes since the NICE proposal is parametrized by a flexible neural network.

We use *ring5* as an example to demonstrate the results. We assume $\pi_0(x) = \mathcal{N}(0, \sigma^2 I)$ as the initial distribution, and optimize $\sigma$ through maximum likelihood. Then we run both methods, and use the resulting particles to estimate $p_d$. As shown in Figures 7a and 7b, HMC fails and there is a large gap between true and estimated statistics. This also explains why the ESS is lower than 1 for HMC for *ring5* in Table 1.

Another reasonable measurement to consider is Gelman's R hat diagnostic [36], which evaluates performance across multiple sampled chains. We evaluate this over the rings5 domain (where the statistics is the distance to the origin), using 32 chains with 5000 samples and 1000 burn-in steps for each sample. HMC gives a R hat value of 1.26, whereas A-NICE-MC gives a R hat value of 1.002 [4]. This suggest that even with 32 chains, HMC does not succeed at estimating the distribution reasonably well.

**Does training increase ESS?**  We show in Figure 8 that in all cases ESS increases with more training iterations and bootstrap rounds, which also indicates that using the pairwise discriminator is effective at reducing autocorrelation.

Admittedly, training introduces an additional computational cost which HMC could utilize to obtain more samples initially (not taking parameter tuning into account), yet the initial cost can be amortized thanks to the improved ESS. For example, in the *ring5* domain, we can reach an ESS of $121.54$ in approximately $550$ seconds (2500 iterations on 1 thread CPU, bootstrap included). If we then sample from the trained A-NICE-MC, it will catch up with HMC in less than 2 seconds.

Next, we demonstrate the effectiveness of A-NICE-MC on Bayesian logistic regression, where the posterior has a single mode in a higher dimensional space, making HMC a strong candidate for the task. However, in order to achieve high ESS, HMC samplers typically use many leap frog steps and require gradients at every step, which is inefficient when $\nabla_x U(x)$ is computationally expensive. A-NICE-MC only requires running $f_\theta$ or $f_\theta^{-1}$ once to obtain a proposal, which is much cheaper computationally. We consider three datasets - *german* (25 covariates, 1000 data points), *heart* (14 covariates, 532 data points) and *australian* (15 covariates, 690 data points) - and evaluate the lowest ESS across all covariates (following the settings in [35]), where we obtain 5000 samples after 1000

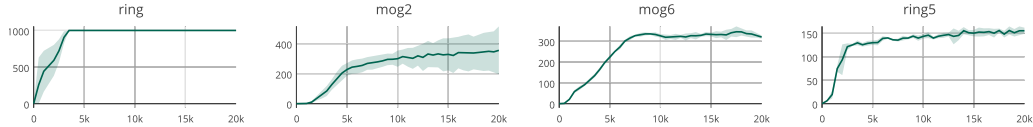

Figure 8: ESS with respect to the number of training iterations.

Table 2: ESS and ESS per second for Bayesian logistic regression tasks.

| ESS | A-NICE-MC | HMC | | ESS/s | A-NICE-MC | HMC |
|---|---|---|---|---|---|---|
| german | 926.49 | **2178.00** | | german | **1289.03** | 216.17 |
| heart | 1251.16 | **5000.00** | | heart | **3204.00** | 1005.03 |
| australian | 1015.75 | **1345.82** | | australian | **1857.37** | 289.11 |

burn-in samples. For HMC we use 40 leap frog steps and tune the step size for the best ESS possible. For A-NICE-MC we use the same hyperparameters for all experiments (details in Appendix E.5). Although HMC outperforms A-NICE-MC in terms of ESS, the NICE proposal is less expensive to compute than the HMC proposal by almost an order of magnitude, which leads to higher ESS *per second* (see Table 2).

## 6 Discussion

To the best of our knowledge, this paper presents the first likelihood-free method to train a parametric MCMC operator with good mixing properties. The resulting Markov Chains can be used to target both empirical and analytic distributions. We showed that using our novel training objective we can leverage flexible neural networks and volume preserving flow models to obtain domain-specific transition kernels. These kernels significantly outperform traditional ones which are based on elegant yet very simple and general-purpose analytic formulas. Our hope is that these ideas will allow us to bridge the gap between MCMC and neural network function approximators, similarly to what "black-box techniques" did in the context of variational inference [1].

Combining the guarantees of MCMC and the expressiveness of neural networks unlocks the potential to perform fast and accurate inference in high-dimensional domains, such as Bayesian neural networks. This would likely require us to gather the initial samples through other methods, such as variational inference, since the chances for untrained proposals to "stumble upon" low energy regions is diminished by the curse of dimensionality. Therefore, it would be interesting to see whether we could bypass the bootstrap process and directly train on $U(x)$ by leveraging the properties of flow models. Another promising future direction is to investigate proposals that can rapidly adapt to changes in the data. One use case is to infer the latent variable of a particular data point, as in variational autoencoders. We believe it should be possible to utilize meta-learning algorithms with data-dependent parametrized proposals.

## Acknowledgements

This research was funded by Intel Corporation, TRI, FLI and NSF grants 1651565, 1522054, 1733686. The authors would like to thank Daniel Lévy for discussions on the NICE proposal proof, Yingzhen Li for suggestions on the training procedure and Aditya Grover for suggestions on the implementation.

## Footnotes

[1]We provide a more rigorous justification in Appendix B.

[2] The cycle consistency loss (as in CycleGAN [31]) introduces a regularization term for this condition; we added this to the REINFORCE objective but were not able to achieve satisfactory results.

[3]The results are not very sensitive to changes in $\gamma$; we also tried Maximum Mean Discrepancy (MMD, see [32] for details) and achieved similar results.

[4]For R hat values, the perfect value is 1, and 1.1-1.2 would be regarded as too high.

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
