[Supplementary Material]

## A  Estimating Effective Sample Size

Assume a target distribution $p(x)$, and a Markov chain Monte Carlo (MCMC) sampler that produces a set of N correlated samples $\{x_i\}_1^N$ from some distribution $q(\{x_i\}_1^N)$ such that $q(x_i) = p(x_i)$. Suppose we are estimating the mean of $p(x)$ through sampling; we assume that increasing the number of samples will reduce the variance of that estimate.

Let $V = \text{Var}_q[\sum_{i=1}^N x_i/N]$ be the variance of the mean estimate through the MCMC samples. The effective sample size (ESS) of $\{x_i\}_1^N$, which we denote as $M = ESS(\{x_i\}_1^N)$, is the number of independent samples from $p(x)$ needed in order to achieve the same variance, i.e. $\text{Var}_p[\sum_{j=1}^M x_j/M] = V$. A practical algorithm to compute the ESS given $\{x_i\}_1^N$ is provided by:

$$ESS(\{x_i\}_1^N) = \frac{N}{1 + 2\sum_{s=1}^{N-1}(1 - \frac{s}{N})\rho_s} \tag{13}$$

where $\rho_s$ denotes the autocorrelation under $q$ of $x$ at lag $s$. We compute the following empirical estimate $\hat{\rho}_s$ for $\rho_s$:

$$\hat{\rho}_s = \frac{1}{\hat{\sigma}^2(N-s)} \sum_{n=s+1}^N (x_n - \hat{\mu})(x_{n-s} - \hat{\mu}) \tag{14}$$

where $\hat{\mu}$ and $\hat{\sigma}$ are the empirical mean and variance obtained by an independent sampler.

Due to the noise in large lags $s$, we adopt the approach of [37] where we truncate the sum over the autocorrelations when the autocorrelation goes below 0.05.

## B  Justifications for Objective in Equation 3

We consider two necessary conditions for $p_d$ to be the stationary distribution of the Markov chain, which can be translated into a new algorithm with better optimization properties, described in Equation 3.

**Proposition 1.** *Consider a sequence of ergodic Markov chains over state space $\mathcal{S}$. Define $\pi_n$ as the stationary distribution for the $n$-th Markov chain, and $\pi_n^t$ as the probability distribution at time step $t$ for the $n$-th chain. If the following two conditions hold:*

1. *$\exists b > 0$ such that the sequence $\{\pi_n^b\}_{n=1}^\infty$ converges to $p_d$ in total variation;*

2. *$\exists \epsilon > 0$, $\rho < 1$ such that $\forall n > 0$ if $\|\pi_n^t - p_d\|_{TV} < \epsilon$, then $\|\pi_n^{t+1} - p_d\|_{TV} < \rho\|\pi_n^t - p_d\|_{TV}$ ;*

*then the sequence of stationary distributions $\{\pi_n\}_{n=1}^\infty$ converges to $p_d$ in total variation.*

*Proof.* The goal is to prove that $\forall \delta > 0, \exists N > 0, T > 0$, such that $\forall n > N, t > T, \|\pi_n^t - p_d\|_{TV} < \delta$.

According to the first assumption, $\exists N > 0$, such that $\forall n > N, \|\pi_n^b - p_d\|_{TV} < \epsilon$.

Therefore, $\forall n > N, \forall \delta > 0, \exists T = b + \max(0, \lceil \log_\rho \delta - \log_\rho \epsilon \rceil) + 1$, such that $\forall t > T$,

$$\|\pi_n^t - p_d\|_{TV}$$
$$= \|\pi_n^b - p_d\|_{TV} \prod_{i=b}^{t-1} \frac{\|\pi_n^{i+1} - p_d\|_{TV}}{\|\pi_n^i - p_d\|_{TV}}$$
$$< \epsilon\rho^{t-b} < \epsilon\rho^{T-b} < \epsilon \cdot \frac{\delta}{\epsilon} = \delta \tag{15}$$

The first inequality uses the fact that $\|\pi_n^b - p_d\|_{TV} < \epsilon$ (from Assumption 1), and $\|\pi_n^{t+1} - p_d\|_{TV}/\|\pi_n^t - p_d\|_{TV} < \rho$ (from Assumption 2). The second inequality is true because $\rho < 1$ by Assumption 2. The third inequality uses the fact that $T - b > \lceil \log_\rho \delta - \log_\rho \epsilon \rceil$ (from definition of $T$), so $\rho^{T-b} < \delta/\epsilon$. Hence the sequence $\{\pi_n\}_{n=1}^\infty$ converges to $p_d$ in total variation. $\square$

Moreover, convergence in total variation distance is equivalent to convergence in Jensen-Shannon (JS) divergence[15], which is what GANs attempt to minimize [4]. This motivates the use of GANs to achieve the two conditions in Proposition 1. This suggests a new optimization criterion, where we look for a $\theta$ that satisfies both conditions in Proposition 1, which translates to Equation 3.

## C  Proof of Theorem 1

*Proof.* For any $(x, v)$ and $(x', v')$, $g$ satisfies:

$$
\begin{aligned}
g(x', v' | x, v) &= \frac{1}{2} \left| \det \frac{\partial f(x, v)}{\partial (x, v)} \right|^{-1} \mathbb{I}(x', v' = f(x, v)) + \frac{1}{2} \left| \det \frac{\partial f(x, v)}{\partial (x, v)} \right| \mathbb{I}(x', v' = f^{-1}(x, v)) \\
&= \frac{1}{2} \mathbb{I}(x', v' = f(x, v)) + \frac{1}{2} \mathbb{I}(x', v' = f^{-1}(x, v)) \\
&= \frac{1}{2} \mathbb{I}(x, v = f^{-1}(x', v')) + \frac{1}{2} \mathbb{I}(x, v = f(x', v')) \\
&= g(x, v | x', v')
\end{aligned}
\tag{16}
$$

where $\mathbb{I}(\cdot)$ is th indicator function, the first equality is the definition of $g(x', v' | x, v)$, the second equality is true since $f(x, v)$ is volume preserving, the third equality is a reparametrization of the conditions, and the last equality uses the definition of $g(x, v | x', v')$ and $f$ is volume preserving, so the determinant of the Jacobian is 1. $\qquad\square$

Theorem 1 allows us to use the ration $p(x', v')/p(x, v)$ when performing the MH step.

## D  Details on the Pairwise Discriminator

Similar to the settings in MGAN objective, we consider two chains to obtain samples:

- Starting from a data point $x$, sample $z_1$ in $B$ steps.
- Starting from some noise $z$, sample $z_2$ in $B$ steps; and from $z_2$ sample $z_3$ in $M$ steps.

For the "generated" (fake) data, we use two type of pairs $(x, z_1)$, and $(z_2, z_3)$. This is illustrated in Figure 9. We assume equal weights between the two types of pairs.

Figure 9: Illustration of the generative process for the pairwise discriminator. We block the gradient for $z_2$ to further parallelize the process and improve training speed.

# E  Additional Experimental Details

## E.1  Architectures for Generative Model for Images

Code is available at https://github.com/ermongroup/markov-chain-gan.

Let 'fc $n$, (activation)' denote a fully connected layer with $n$ neurons. Let 'conv2d $n$, $k$, $s$, (activation)' denote a convolutional layer with $n$ filters of size $k$ and stride $s$. Let 'deconv2d $n$, $k$, $s$, (activation)' denote a transposed convolutional layer with $n$ filters of size $k$ and stride $s$.

We use the following model to generate Figure 1 (MNIST).

| encoder | decoder | discriminator |
|---|---|---|
| fc 600, lrelu | fc 600, lrelu | conv2d 64, $4 \times 4$, $2 \times 2$, relu |
| fc 100, linear | fc 784, sigmoid | conv2d 128, $4 \times 4$, $2 \times 2$, lrelu |
| | | fc 600, lrelu |
| | | fc 1, linear |

We use the following model to generate Figure 3 (CelebA, top)

| encoder | decoder | discriminator |
|---|---|---|
| conv2d 64, $4 \times 4$, $2 \times 2$, lrelu | fc $16 \times 16 \times 64$, lrelu | conv2d 64, $4 \times 4$, $2 \times 2$, relu |
| fc 200, linear | deconv2d 3, $4 \times 4$, $2 \times 2$, tanh | conv2d 128, $4 \times 4$, $2 \times 2$, lrelu |
| | | conv2d 256, $4 \times 4$, $2 \times 2$, lrelu |
| | | fc 1, linear |

For the bottom figure in Figure 3, we add a residual connection such that the input to the second layer of the decoder is the sum of the outputs from the first layers of the decoder and encoder (both have shape $16 \times 16 \times 64$); we add a highway connection from input image to the output of the decoder:

$$\bar{x} = \alpha x + (1 - \alpha)\hat{x}$$

where $\bar{x}$ is the output of the function, $\hat{x}$ is the output of the decoder, and $\alpha$ is an additional transposed convolutional output layer with sigmoid activation that has the same dimension as $\hat{x}$.

We use the following model to generate Figure 5 (CelebA, pairwise):

| encoder | decoder | discriminator |
|---|---|---|
| conv2d 64, $4 \times 4$, $2 \times 2$, lrelu | fc 1024, relu | conv2d 64, $4 \times 4$, $2 \times 2$, relu |
| conv2d 64, $4 \times 4$, $2 \times 2$ | fc $8 \times 8 \times 128$, relu | conv2d 128, $4 \times 4$, $2 \times 2$, lrelu |
| fc 1024, lrelu | deconv2d 64, $4 \times 4$, $2 \times 2$, relu | conv2d 256, $4 \times 4$, $2 \times 2$, lrelu |
| fc 200 linear | deconv2d 3, $4 \times 4$, $2 \times 2$, tanh | fc 1, linear |

For the pairwise discriminator, we double the number of filters in each convolutional layer. According to [17], we only use batch normalization in the generator for all experiments.

## E.2  Analytic Forms of Energy Functions

Let $f(x|\mu, \sigma)$ denote the log pdf of $\mathcal{N}(\mu, \sigma^2)$.

The analytic form of $U(x)$ for *ring* is:

$$U(x) = \frac{(\sqrt{x_1^2 + x_2^2} - 2)^2}{0.32} \tag{17}$$

The analytic form of $U(x)$ for *mog2* is:

$$U(x) = f(x|\mu_1, \sigma_1) + f(x|\mu_2, \sigma_2) - \log 2 \tag{18}$$

where $\mu_1 = [5, 0]$, $\mu_2 = [-5, 0]$, $\sigma_1 = \sigma_2 = [0.5, 0.5]$.

The analytic form of $U(x)$ for *mog6* is:

$$U(x) = \sum_{i=1}^{6} f(x|\mu_i, \sigma_i) - \log 6 \tag{19}$$

where $\mu_i = [\sin \frac{i\pi}{3}, \cos \frac{i\pi}{3}]$ and $\sigma_i = [0.5, 0.5]$.

The analytic form of $U(x)$ for *ring5* is:

$$U(x) = \min(u_1, u_2, u_3, u_4, u_5) \tag{20}$$

where $u_i = (\sqrt{x_1^2 + x_2^2} - i)^2/0.04$.

### E.3 Benchmarking Running Time

Since the runtime results depends on the type of machine, language, and low-level optimizations, we try to make a fair comparison between HMC and A-NICE-MC on TensorFlow [38].

Our code is written and executed in TensorFlow 1.0. Due to the optimization of the computation graphs in TensorFlow, the wall-clock time does not seem to be exactly linear in some cases, even when we force the program to use only 1 thread on the CPU. The wall-clock time is affected by 2 aspects, batch size and number of steps. We find that the wall-clock time is relatively linear with respect to the number of steps, and not exactly linear with respect to the batch size.

Given a fixed number of steps, the wall-clock time is constant when the batch size is lower than a threshold, and then increases approximately linearly. To perform speed benchmarking on the methods, we select the batch size to be the value around the threshold, in order to prevent significant under-estimates of the efficiency.

We found that the graph is much more optimized if the batch size is determined before execution. Therefore, we perform all the benchmarks on the optimized graph where we specify a batch size prior to running the graph. For the energy functions, we use a batch size of 2000; for Bayesian logistic regression we use a batch size of 64.

### E.4 Hyperparameters for the Energy Function Experiments

For all the experiments, we use same hyperparameters for both A-NICE-MC and HMC. We sample $x_0 \sim \mathcal{N}(0, I)$ and run the chain for 1000 burn-in steps and evaluate the samples from the next 1000 steps.

For HMC we use 40 leapfrog steps and a step size of 0.1. For A-NICE-MC we consider $f_\theta(x, v)$ with three coupling layers, which updates $v$, $x$ and $v$ respectively. The motivation behind this particular architecture is to ensure that both $x$ and $v$ could affect the updates to $x'$ and $v'$. In each coupling layer, we select the function $m(\cdot)$ to be a one-layer NN with 400 neurons. The discriminator is a three layer MLP with 400 neurons each. Similar to the settings in Section 3.1, we use the gradient penalty method in [17] to train our model.

For bootstrapping, we first collect samples by running the NICE proposal over the untrained $f_\theta$, and for every 500 iterations we replace half of the samples with samples from the latest trained model. All the models are trained with AdaM [39] for 20000 iterations with $B = 4$, $M = 2$, batch size of 32 and learning rate of $10^{-4}$.

### E.5 Hyperparameters for the Bayesian Logistic Regression Experiments

For HMC we tuned the step size parameter to achieve the best ESS possible on each dataset, which is 0.005 for *german*, 0.01 for *heart* and 0.0115 for *australian* (HMC performance on *australian* is extremely sensitive to the step size). For A-NICE-MC we consider $f(x, v)$ with three coupling layers, which updates $v$, $x$ and $v$ respectively; we set $v$ to have 50 dimensions in all the experiments. $m(\cdot)$ is a one-layer NN with 400 neurons for the top and bottom coupling layer, and a two-layer NN with 400 neurons each for the middle layer. The discriminator is a three layer MLP with 800 neurons each. We use the same training and bootstrapping strategy as in Appendix E.4. All the models are trained with AdaM for 20000 iterations with $B = 16$, $M = 2$, batch size of 32 and learning rate of $5 \times 10^{-4}$.

## E.6 Architecture Details

The following figure illustrates the architecture details of $f_\theta(x, v)$ for A-NICE-MC experiments. We do not use batch normalization (or other normalization techniques), since it slows the execution of the network and does not provide much ESS improvement.

(a) NICE architecture for energy functions.    (b) NICE architecture for Bayesian logistic regression.

# F   Extended Images

We only displayed a small number of images in the main text due to limited space. Here we include the extended version of images for our image generation experiments.

## F.1   Extended Images for Figure 1

Figure 11: Samples from $\pi_1$ to $\pi_{50}$ from a model trained on the MNIST dataset. Each row are samples from the same chain.

## F.2  Extended Images for Figure 3

The following models are trained with the original MGAN objective (without pairwise discriminator).

Figure 12: Samples from $\pi_1$ to $\pi_{50}$ from a model trained on the CelebA dataset. Each row are samples from the same chain.

Figure 13: Samples from $\pi_1$ to $\pi_{50}$ from a model trained on the CelebA dataset, where the model has shortcut connections. Each row are samples from the same chain.

## F.3 Extended Images for Figure 5

The following images are trained on the same model with shortcut connections.

Figure 14: Samples from $\pi_1$ to $\pi_{50}$ from a model trained on the CelebA dataset without pairwise discriminator. Each row are samples from the same chain.

Figure 15: Samples from $\pi_1$ to $\pi_{50}$ from a model trained on the CelebA dataset with pairwise discriminator. Each row are samples from the same chain.