[Reviews · NeurIPS 2017]

Reviewer 1



This paper describes a novel adversarial training procedure to fit a generative model described by a Markov chain to sampled data. The Markov chain transitions are based on NICE, and so are reversible and volume preserving. It is therefore straightforward to use these as proposals in a Metropolis MCMC method to sample from arbitrary distributions. By repeatedly fitting the Markov chain model to samples from preliminary runs, we can hope that we'll end up with an MCMC method that mixes well on an arbitrary target distribution. Like HMC, the Markov chain is actually on a joint distribution of the parameters of interest, and some auxiliary random draws used to make a deterministic proposal. It's a promising idea. The MCMC examples in the paper aren't particularly compelling. They're ok, but limited. However, this is the sort of algorithm that could work and be useful in some realistic situations. Quality The paper appears to be technically sound. It describes a valid MCMC method. It's unclear when a bootstrapping procedure (a precise procedure isn't given in the paper) will converge to a working proposal mechanism. The paper ends alluding to this issue. Variational inference would find a feasible region of parameters, but (like all MCMC methods), the method could be trapped there. There is a danger that parts of the distribution are ignored. The cost function doesn't ensure that the proposal mechanism will get everywhere. Future work could investigate that further. Mixing updates with a more standard MCMC mechanism may reveal blind spots. The toy experiments indicate that the method can learn to hop modes with similar densities in 2D. But don't reflect statistical models where MCMC inference is used. Of course section 3 shows some interesting transitions can be learned in high dimensions. However, the interest is in application to MCMC, as there is plenty of work on generating synthetic images using a variety of procedures. The experiments on Bayesian logistic regression are a starting point for looking at real statistical analyses. Personally, if forced to use HMC on logistic regression, I would estimate and factor the covariance of the posterior S = LL', using a preliminary run, or other approximate inference. Then reparameterize the space z = L^{-1}x, and run HMC on z. Equivalently set the mass matrix to be S^{-1}. Learning a linear transformation is simpler and more standard than fitting neural nets -- would the added complication still be worth it? Projects like Stan come with demonstrations. Are there more complicated distributions from real problems where A-NICE-MC would readily out-perform current practice? Clarity The paper is fairly clearly written. The MCMC algorithm isn't fully spelled out, or made concrete (what's p(v)?). It would only be reproducible (to some extent) for those familiar with both HMC and NICE. Originality While the general idea of fitting models and adapting proposals has appeared periodically in the MCMC literature, this work is novel. The adversarial method proposed seems a good fit for learning proposals, rather than boldly trying to learn an explicit representation of the global distribution. Significance Statisticians are unlikely to adopt this work in its current form. It's a rather heavier dependency than "black-box variational inference", and I would argue that fitting neural nets is not generically black-box yet. The work is promising, but more compelling statistical examples would help. Volume preserving proposals won't solve everything. One might naively think that the only challenge for MCMC is to come up with proposal operators that can make proposals to all the modes. However, typical samples from the posterior of a rich model can potentially vary in density a lot. There could be narrow tall modes, or short large basins of density, potentially with comparable posterior mass. Hierarchical models in particular can have explanations with quite different densities. A volume-preserving proposal can usually only change the log-density by ~1, or the Metropolis rule will reject too often. Minor The discussion of HMC and rules that return to (x,v) after two updates are missing the idea of negating the velocity. In HMC, the dynamics only reverse if we deterministically set v' <- -v' after the first update. Please don't use the term "Exact Sampling" in the header to section 4.1. In the MCMC literature that term usually means generating perfect independent samples from a target distribution, a much stronger requirement than creating updates that leave the target distribution invariant: http://dbwilson.com/exact/ The filesize of this paper is larger than it should be, making the paper slow to download and display. Figure 2 seems to be stored as a .jpeg, but should be a vector graphic -- or if you can't manage that, a .png that hasn't passed through being a .jpeg. Figure 3 doesn't seem to be stored as a .jpeg, and I think would be ~10x smaller if it were(?). Several bibtex entries aren't capitalized properly. Protect capital letters with {.}.

Reviewer 2



I have read the author feedback; thanks for the comments and clarifications. ---- This is a fun paper, which proposes a new way to train Markov operators whose stationary distribution matches a desired target. Two applications are considered: one, as a manner for generating synthetic data with a Markov chain whose stationary distribution is a given training data distribution, and two, as a means for learning efficient MCMC kernels for Bayesian posterior inference. The basic idea is to train a generative model and a discriminator according to a GAN objective, where the generative model is parameterized by repeated application of a transition kernel to samples from some initial distribution. I think that this is a great way to leverage Markov chain properties for constructing generative networks. I expect the choice of objective function in eq (3) is critical to making this work in practice: as the paper states, backpropagation through a long Markov chain, while technically possible, would be prohibitively expensive. Instead, the objective function is chosen to roughly encourage both fast convergence to the ergodic regime, and that the target distribution remains invariant to repeated sampling from the transition distribution. The results here look visually quite good, and this is a much cleaner approach overall than e.g. the earlier (cited) generative stochastic networks. I find the application to sampling from Bayesian posterior distributions slightly less convincing, but I think the results are nice and there are some good ideas presented. In particular, the connection between volume-preserving flows and volume preservation required for symmetric MH proposals is quite clever, and I think the NICE proposal (or something quite similar) could find other application. My skepticism stems from: • What is the real computational cost of training? I find it hard to believe that in real-world problems it is more efficient to both train the NICE proposal as well as run the MCMC algorithm, than it would be to simply go run the HMC algorithm for longer. Relatedly, • how well does the bootstrap procedure in section 4.5 actually work? How do you choose the initial parameters \theta_0, and how sensitive is this to poor initialization? A pathological choice of initial operator could prevent ever finding anything resembling “real” samples from p_d. In general this sort of “adaptive” MCMC algorithm which updates the parameter simultaneously while performing inference is delicate, and is not necessarily ergodic, losing the guarantees that come with static MCMC algorithms. See e.g. Roberts & Rosenthal “Ergodicity of Adaptive MCMC” for algorithms, theory, and a discussion of the challenges. • While HMC certainly would fail to move between the modes of mog2, mog6, and ring5, I would point out that these are not really good examples of the sort of distributions which actually arise as Bayesian posteriors. (The logistic regression example is better on that account, but I would be cautious using Bayesian logistic regression as a benchmark for sampler performance; see the paper by Chopin and Ridgeway “Leave Pima Indians alone: binary regression as a benchmark for Bayesian computation”.) Minor comments: • The phrase NICE is used throughout, starting with the title and abstract, but the acronym is not finally defined until the middle of page 5! • Consider using additional sampling diagnostics than ESS, and consider an HMC benchmark which averages over multiple chains. Using multiple chains (and looking at Gelman’s R-hat diagnostic criteria) should yield better results for HMC on the synthetic data examples, as the algorithm would not collapse into a single mode.